# Circulating Tumor DNA Reflects Uveal Melanoma Responses to Protein Kinase C Inhibition

**DOI:** 10.3390/cancers13071740

**Published:** 2021-04-06

**Authors:** John J. Park, Russell J. Diefenbach, Natalie Byrne, Georgina V. Long, Richard A. Scolyer, Elin S. Gray, Matteo S. Carlino, Helen Rizos

**Affiliations:** 1Department of Biomedical Sciences, Faculty of Medicine, Health and Human Sciences, Macquarie University, Sydney, NSW 2109, Australia; john.park4@hdr.mq.edu.au (J.J.P.); russell.diefenbach@mq.edu.au (R.J.D.); 2Melanoma Institute Australia, The University of Sydney, Sydney, NSW 2065, Australia; georgina.long@sydney.edu.au (G.V.L.); richard.scolyer@health.nsw.gov.au (R.A.S.); matteo.carlino@sydney.edu.au (M.S.C.); 3Department of Medical Oncology, Westmead and Blacktown Hospitals, Sydney, NSW 2145, Australia; natalie.byrne@sydney.edu.au; 4Department of Medical Oncology, Royal North Shore Hospital and Mater Hospitals, Sydney, NSW 2065, Australia; 5Faculty of Medicine and Health, The University of Sydney, Sydney, NSW 2006, Australia; 6Tissue Pathology and Diagnostic Oncology, Royal Prince Alfred Hospital and NSW Health Pathology, Sydney, NSW 2050, Australia; 7Centre for Precision Health, Edith Cowan University, Joondalup, WA 6027, Australia; e.gray@ecu.edu.au

**Keywords:** uveal melanoma, circulating tumor DNA, next generation sequencing, PKC inhibitor, liquid biopsy, treatment, response, melanoma

## Abstract

**Simple Summary:**

Uveal melanoma (UM) is a rare cancer, with no effective standard systemic therapy in the metastatic setting. Over 95% of UM harbor activating driver mutations that can be detected in the circulation. In this study, circulating tumor DNA (ctDNA) was measured in 17 metastatic UM patients treated with protein kinase C inhibitor (PKCi)-based therapy. ctDNA predicted response to targeted therapy and increasing UM ctDNA preceded radiological progression with a lead-time of 4–10 weeks. Next generation sequencing (NGS) of ctDNA also identified prognostic and treatment resistance mutations. Longitudinal ctDNA monitoring is useful for monitoring disease response and progression in metastatic UM and is a valuable addition to adaptive clinical trial design.

**Abstract:**

The prognosis for patients with UM is poor, and recent clinical trials have failed to prolong overall survival (OS) of these patients. Over 95% of UM harbor activating driver mutations, and this allows for the investigation of ctDNA. In this study, we investigated the value of ctDNA for adaptive clinical trial design in metastatic UM. Longitudinal plasma samples were analyzed for ctDNA in 17 metastatic UM patients treated with PKCi-based therapy in a phase 1 clinical trial setting. Plasma ctDNA was assessed using digital droplet PCR (ddPCR) and a custom melanoma gene panel for targeted next generation sequencing (NGS). Baseline ctDNA strongly correlated with baseline lactate dehydrogenase (LDH) (*p* < 0.001) and baseline disease burden (*p* = 0.002). Early during treatment (EDT) ctDNA accurately predicted patients with clinical benefit to PKCi using receiver operator characteristic (ROC) curves (AUC 0.84, [95% confidence interval 0.65–1.0, *p* = 0.026]). Longitudinal ctDNA assessment was informative for establishing clinical benefit and detecting disease progression with 7/8 (88%) of patients showing a rise in ctDNA and targeted NGS of ctDNA revealed putative resistance mechanisms prior to radiological progression. The inclusion of longitudinal ctDNA monitoring in metastatic UM can advance adaptive clinical trial design.

## 1. Introduction

Uveal melanoma (UM) is the most common primary intraocular malignancy [1]. The tumor arises from melanocytes within the uveal tract, with more than 90% of cases involving the choroid followed by iris and ciliary body [2]. UM is a rare cancer, affecting approximately 5–7 individuals per million each year [1,3,4,5]. Despite successful local treatment with either surgery or radiation therapy, approximately 50% of patients with UM will develop metastatic disease [6] with over 90% of metastases occurring in the liver [7]. Currently there is no effective systemic treatment in metastatic UM and the median progression free survival (PFS) and overall survival (OS) are 3.3 months and 10.2 months, respectively [8].

Nearly 95% of UM harbor mutually exclusive activating driver mutations in *GNAQ*, *GNA11*, *CYSTLR2* and *PLCβ4* genes [9,10,11,12,13,14]. Molecular profiling, cytogenetic and transcriptomic analysis of UM have provided accurate prognostic information [9,15]. Additional hot spot mutations affecting the *EIF1AX* and *SF3B1* genes are associated with better prognosis whereas loss of function *BAP1* gene alterations are correlated with the development of UM metastases and poor prognosis [9,16]. Somatic copy number alterations such as loss of chromosome 3, 6q and 8q are also associated with poor prognosis [17]. The specific and defined mutation profile of UM provides an excellent opportunity to investigate the utility of circulating tumor DNA (ctDNA) as a biomarker to detect the presence of metastatic disease and to rapidly monitor response to early-phase drug therapies.

In cutaneous melanoma (CM), baseline ctDNA is strongly correlated with tumor burden in patients with advanced stage disease [18] and is associated with overall response rate and PFS in patients treated with targeted therapies [18,19]. A decline in ctDNA within 8 weeks of treatment initiation also predicts response to both combined BRAF and MEK inhibition and immunotherapy in CM [19,20]. In metastatic UM, ctDNA levels correlate with tumor burden and the presence of liver metastases [21,22] and are also prognostic for PFS and OS [21]. The value of ctDNA in monitoring and predicting response to trial drug therapies has not, to the best of our knowledge, been previously investigated. This is particularly relevant in metastatic UM as there are currently no effective systemic treatments, but significant ongoing clinical trial activity evaluating novel therapies. In this study, we sought to assess ctDNA in metastatic UM patients treated with protein kinase C inhibitor (PKCi)-based therapy in a phase 1 clinical trial setting (NCT02601378). Using two methods, droplet digital PCR (ddPCR) and targeted Ion Torrent next generation sequencing (NGS), we evaluated the utility of plasma ctDNA in monitoring and predicting clinical outcomes including best response to therapy and PFS.

## 2. Materials and Methods

### 2.1. Patients and Treatment

Seventeen patients with metastatic UM with known mutations in GNAQ, GNA11 and CYSTLR2, treated with the novel PKCi, LXS196 (*n* = 17) at Westmead Hospital, Sydney, Australia as part of an experimental dose escalation phase 1 clinical trial between November 2016 to August 2018 were included in this study. Written consent was obtained from all patients with metastatic UM under approved Human Research ethics committee protocols from Royal Prince Alfred Hospital (Protocol X15-0454 and HREC/11/RPAH/444).

### 2.2. Patient and Disease Characteristics and Response Assessment

Patient demographics and clinicopathologic features including mutation status, Eastern Cooperative Oncology Group (ECOG) performance status, and baseline LDH levels (units/litre; U/L) were collected. Baseline disease burden was determined by the sum of the product of bi-dimensional diameters (SPOD) for every metastasis ≥5 mm in the long axis (≥15 mm in the short axis for lymph nodes). Investigator-determined objective responses were assessed radiologically with computed tomography (CT) scans at two monthly intervals using Response Evaluation Criteria in Solid Tumors (RECIST) 1.1 criteria [23]. Clinical progression was defined by primary clinician’s assessment of disease progression in patients without re-staging imaging and were classified as progressive disease (PD). Clinical benefit was defined by patients who had partial response (PR) or stable disease (SD) for ≥6 months.

### 2.3. Plasma Preparation

Plasma samples were collected at baseline (prior to therapy start), EDT (early during treatment between 14–30 days of commencing PKCi-based therapy) and at later time points during therapy (on-treatment samples). PROG samples were defined as plasma samples taken within 30 days (before or after) of disease progression confirmed by imaging or clinical progression as determined by the treating clinician. NGS analysis was performed on baseline plasma samples and on the last available on-treatment plasma sample. Blood (10 mL) was collected in EDTA tubes (Becton Dickinson, Franklin Lakes, NJ, USA) and processed within 4 h from blood draw. Tubes were spun at 800 g for 15 min for plasma collection, followed by a second centrifugation at 1600 g for 10 min to remove cellular debris. Plasma was stored in 1–2 mL aliquots at −80 °C.

### 2.4. Purification of Circulating Free DNA from Plasma

Plasma circulating free DNA was extracted using the QIAamp Circulating Nucleic Acid Kit (Qiagen, Hilden, Germany) according to the manufacturer’s instructions. Circulating free DNA was purified from 1–4 mL of plasma and the final elution volume was 25 μL. Total circulating free DNA was quantified using a Qubit dsDNA high sensitivity assay kit and a Qubit fluorometer 3 (Life Technologies, Carlsbad, CA, USA) according to the manufacturer’s instructions.

### 2.5. ddPCR Analysis of ctDNA from Plasma

The copy number of ctDNA per milliliter of plasma was determined using the QX200 ddPCR system (Bio-Rad, Hercules, CA, USA), as previously described [20]. Commercially available (GNAQ Q209P and GNA11 Q209L; Bio-Rad) and customized probes [22] (GNAQ R183C and CYSTLR2 L129Q) were used to analyze ctDNA by ddPCR. The DNA copy number/mL of plasma for mutant and wild-type circulating DNA species was determined with QuantaSoft software version 1.7.4 (Bio-Rad, Hercules, CA, USA) using a manual threshold setting. If analysis confirmed only 1 positive ctDNA mutant copy per 20 μL, the ddPCR amplification was repeated up to three times, and the plasma sample was considered positive if ctDNA was positive in at least two repeat experiments. ddPCR results are reported as ctDNA copies/mL.

### 2.6. Custom Melanoma Gene Panel for Targeted NGS of Circulating Free DNA

An Ion Ampliseq HD made-to-order melanoma gene panel was obtained from Life Technologies (Carlsbad, CA, USA). The panel, which consists of 123 amplicons and covers melanoma-associated mutations in 30 gene targets, has been described previously [24]. This melanoma gene panel does not cover the *BAP1* gene. DNA target amplification, using 20 ng circulating free DNA as template, library construction and sequencing were performed as previously described [24]. Ion Torrent NGS results in our study are reported in mutant allele frequency (MAF).

### 2.7. Statistical Analysis

The Spearman rank correlation coefficient was used to test the correlation between the ctDNA copies, and the baseline LDH level, baseline SPOD, or longest diameter of liver metastatic lesion. Kruskal–Wallis test with Dunn’s multiple comparison test was used to compare ctDNA copies in the clinical benefit group and no clinical benefit group. EDT ctDNA copies to predict clinical benefit was measured using Receiver Operating Characteristics (ROC) analysis. Statistical analyses were carried out using GraphPad Prism 9. Positive predictive value for EDT > 16.35 copies/mL was calculated using the following formula: Number of patients showing no clinical benefit with EDT ctDNA > 16.35 copies/mL divided by number of patients with EDT ctDNA > 16.35 copies/mL. Negative predictive value was determined as follows: Number of responding patients with EDT ctDNA ≤ 16.35 copies/mL divided by number of patients with EDT ctDNA ≤ 16.35 copies/mL.

## 3. Results

### 3.1. Patient Characteristics

Seventeen patients with metastatic UM were included in this study; 11 patients received PKCi alone and six patients received PKCi in combination with the human homolog of double minute 2 (HDM2) inhibitor (HDM201). Median follow-up duration was 20.1 weeks (range 6.3–66.0 weeks). Baseline demographic data are detailed in Table 1. The median age was 56 years and the majority of patients were male (10/17; 59%) with an ECOG status of 0 (13/17; 76%). All patients had an established UM driver mutation (GNAQ Q209P (35%), GNA11 Q209L (47%), GNAQ R183Q (12%) and CYSTLR2 L129Q (6%)), and metastatic disease involving the liver. On commencement of the treatment, 11 (65%) patients had elevated LDH levels and 13 (76%) had prior systemic treatment. The majority of patients (12/17; 70%) had a choroidal primary UM, 1 (6%) had an iris primary tumor and for the remaining 4 (24%) patients, the additional component of the primary tumor was unknown. Overall PFS was 3.8 months. Patients with RECIST 1.1 PR (2/17; 12%) or SD ≥ 6 months (4/17; 24%) were classified as the ‘clinical benefit’ group, while patients with SD < 6 months (7/17; 41%) or PD (4/17; 24%) were classified as having ‘no clinical benefit’ group (Appendix A).

### 3.2. Baseline ctDNA Levels Are Associated with Tumor Volume and LDH Level

ctDNA was detected by ddPCR in 16/17 (94%) patients prior to commencing therapy. Median ctDNA was 157.7 copies/mL with a range of 0–7172 copies/mL. Baseline ctDNA was strongly correlated with baseline LDH (Spearman’s rank r = 0.7941, *p* < 0.001) and baseline SPOD (Spearman’s rank r = 0.7206, *p* = 0.002) (Figure 1A,B). As expected, the total lesion SPOD was significantly correlated to liver SPOD in these patients (Spearman’s rank r = 0.8676, *p* < 0.01; Appendix A); however, the baseline ctDNA did not correlate with the longest diameter of liver lesion (Spearman’s rank r = 0.4027, *p* = 0.110) or liver SPOD (*p* = 0.06, Spearman’s rank r = 0.4632) (Figure 1C,D). The discrepant correlation between ctDNA versus total lesion SPOD and ctDNA versus liver SPOD was influenced by the distribution of melanoma metastases in patient #6. This patient had multiple disease sites, a relatively high overall tumor burden (6022 mm^2^), but very low liver disease (469 mm^2^) (Appendix A). When patient #6 was excluded, ctDNA levels were significantly correlated with the longest diameter of liver lesion (Spearman’s rank r = 0.6455, *p* < 0.01) and liver SPOD (Spearman’s rank r = 0.7176, *p* < 0.01).

We identified six patients in the clinical benefit group (PR, or SD ≥ 6 months; including 5/6 (83%) patients treated with PKCi monotherapy) and eleven patients in the no clinical benefit group (SD < 6 months or PD; including 6/11 (55%) patients treated with PKCi monotherapy). Baseline ctDNA, SPOD and LDH were compared in the clinical benefit versus no clinical benefit patient groups. Lower median baseline ctDNA was observed in the clinical benefit group (33.8 copies/mL, range 0–333 copies/mL) compared to patients in the no clinical benefit group (196.2 copies/mL, range 15–7172 copies/mL); however, this difference was not statistically significant (Appendix A). Similarly, LDH, median total SPOD, longest diameter of liver lesion and LDH and liver SPOD were lower in the clinical benefit versus no clinical benefit subset; however, these differences were not significantly different (Appendix A). The sample set was too small for multivariate analysis comparing baseline ctDNA, LDH and SPOD to best response.

### 3.3. Prognostic Value of Early during Treatment (EDT) ctDNA

Paired baseline and EDT ctDNA samples were available for 16 patients (patient #15 did not have an EDT sample). Four patients (4/16; 25%) displayed undetectable ctDNA at EDT and three of these patients benefited from PKCi-based therapy (1 with PR and 2 with SD ≥ 6 months; SD for 9.6 and 13.1 months, respectively) and the fourth patient initially had SD but progressed at 3.7 months. Of these four patients, one (patient #5) had low disease volume, undetectable ctDNA at baseline and EDT, SD ≥ 6 months and a PFS of 13.1 months. Of the remaining three patients, ctDNA zero converted at EDT from baseline ctDNA levels ranging from 13–30 ctDNA copies/mL (Figure 2A). All four had a GNA11 Q209L mutation, a below median tumor burden and an LDH level below the upper limit of normal.

Another nine patients (9/12; 75%) had positive ctDNA at baseline and showed a substantial reduction but still detectable ctDNA at EDT (ctDNA reduction range 46–99%). Three of these nine patients (33%) benefited from PKCi; patient #16 achieved PR with delayed zero-conversion of ctDNA at day 57, patient #4 achieved SD ≥ 6 months and showed undetectable ctDNA 41 days post treatment start and patient #1 achieved SD ≥ 6 without zero-conversion of ctDNA. Six patients with reduced, but not undetectable ctDNA levels at EDT had no clinical benefit, with three patients showing SD < 6 months (patients #11, #13 and #14) and three patients with PD (patients #6, #8 and #10) as best response (Figure 2A, Appendix A). The remaining three patients (patients #7, #12, #17) showed an increase in ctDNA from baseline to EDT and all three of these patients did not benefit from therapy (2 with SD < 6 months and 1 with, PD).

EDT ctDNA levels were significantly lower in the clinical benefit patients compared to the no clinical benefit subgroup (*p* = 0.023; Figure 2A). There was no statistical difference in the baseline ctDNA level of the clinical benefit and no clinical benefit group. The changes in baseline ctDNA to EDT ctDNA in both the clinical benefit group and no clinical benefit group were also not statistically significant. The predictive accuracy of ctDNA was also examined using receiver operator characteristic classification (ROC) curves. EDT ctDNA, but not PRE ctDNA or change from PRE to EDT, accurately predicted clinical benefit to PKCi based therapy (AUC 0.84, [95% confidence interval, 0.65–1.0, *p* = 0.026]) (Figure 2B). Based on ROC curve analysis, the sensitivity and specificity for ctDNA > 16.35 copies/mL at EDT in the no clinical benefit group were 70% and 100%, respectively. The positive and negative predictive values for ctDNA > 16.35 copies/mL were 100% and 67%, respectively.

### 3.4. Longitudinal ctDNA Monitoring and Disease Progression

Monitoring ctDNA levels over time was also informative for establishing clinical benefit and detecting disease progression. Collectively, six patients had undetectable ctDNA during treatment (patient #2, #3, #4, #5, #9 and #16) and five of these patients (5/6; 83%) benefited from PKCi-based therapy (PR or SD ≥ 6 months) (Figure 3A). Patient #9 was the only patient with no clinical benefit who had an undetectable ctDNA at EDT and multiple later time points (Figure 3B, Appendix A). Conversely, of the seven patients with consistently detectable ctDNA during therapy (patient #1, #7, #11, #13, #14, #15, #17) (Figure 3, Appendix A) only one patient (patient #1, PFS of 7.4 months; Figure 3C) benefited from therapy and this patient showed a 74% reduction in ctDNA level from baseline to EDT.

Eight out of seventeen patients also had ctDNA samples assessed within 30 days (before or after) of disease progression. In total, 7/8 patients (patient #1, #3 #7, #9, #11, #13 and #15; Figure 3, Appendix A) showed increasing ctDNA prior to radiological confirmation of disease progression with an increase in size of target lesions and new metastases as per RECIST 1.1 (Figure 4). Only patient #17 showed ctDNA levels near progression that were lower than EDT ctDNA despite CT imaging confirming disease progression (Figure 3).

### 3.5. Detection of Driver and Additional Mutations through Ion Torrent NGS

We next examined the ctDNA of these patients using a targeted NGS panel that included gene alterations shown to be prognostic in UM (Table 2). Paired baseline and on-treatment samples (time from baseline to on-treatment sample 0.9–11.3 months) from 17 patients were sequenced and the driver GNAQ, GNA11 and CYSLTR2 mutations identified in the tumor were confirmed by NGS in the baseline and/or on-treatment ctDNA samples in 16/17 patients. The allele frequency of tumor-associated mutations determined by ddPCR and NGS was highly correlated (Spearman’s rank r = 0.968, *p* < 0.001; Appendix A). The GNA11 Q209L driver mutation present in the UM of patient #9 was not detected in the baseline or on-treatment liquid biopsy samples using NGS, and this patient had the lowest tumor burden (SPOD = 200 mm^2^), although not the lowest baseline ctDNA levels by ddPCR (24 copies/mL plasma).

In addition to the UM driver mutations, the hotspot SF3B1 R625 mutation was detected in baseline and on-treatment ctDNA samples derived from four patients with GNAQ or GNA11 driver mutations (patients #3, #6, #7 and #14; Table 2). The allele frequencies of the SF3B1 and driver GNAQ/GNA11 mutations were highly correlated in these eight ctDNA samples (Appendix A). Of the four patients with SF3B1 mutations, one achieved SD ≥ 6 months, two had SD < 6 months and one patient had PD. The median PFS of these SF3B1 mutation-positive patients was 4.6 months, slightly longer than the median PFS of 3.8 months for the whole cohort.

We also identified TP53 mutations in the baseline and/or on-treatment plasma samples of 10 patients and many of these TP53 mutations are established cancer-associated mutations (e.g., S215G, Y220C, G244D, G245S, R282P, R248P/Q/G) [25] (Appendix A). Of these 10 patients, five received combination PKCi+HDM2i and there was an enrichment of TP53 mutations in the on-treatment plasma samples from patients treated with PKCi+HDM2i compared to patients treated with PKCi monotherapy (Fisher exact test, *p* = 0.035). Interestingly, in four of five PKCi+HDM2i patients the TP53 variants were not identified pre-treatment, suggesting the possibility of selection during treatment. It is important to note that many TP53 mutations were detected at low allele frequencies that were below the 0.2% limit of detection of our NGS assay [24] and thus would require further validation.

## 4. Discussion

Currently, there is no effective systemic therapy for metastatic UM and recent clinical trials with targeted therapies and immune checkpoint inhibitors appear not to improve the OS of patients with metastatic UM [26,27]. Numerous phase 1 clinical trials are currently underway including with PKCi-based therapy. In this study, we explored the utility of ctDNA as an early marker of Phase I drug efficacy and resistance in metastatic UM [28].

We noted a strong correlation between baseline UM ctDNA levels and prognostic markers including tumor burden and LDH levels, and this is consistent with previous reports showing that elevated ctDNA reflects higher disease burden and is associated with poor prognosis in UM and CM [20,21]. It is well established that ctDNA is also associated with response, PFS and OS in metastatic CM patients treated with targeted therapies and anti-PD1-based immunotherapies [19,20]. In this study, we explored the utility of ctDNA in monitoring and predicting UM response to PKCi-based therapy. Only one other study has examined longitudinal UM ctDNA levels and treatment response. The latter was a proof-of-concept study including only three UM patients, and although it confirmed that EDT ctDNA was predictive of an anti-PD-1 inhibitor response in a cohort that included various cancer types [29], the UM patients failed to respond to PD-1 inhibitor blockade and ctDNA was only detected in two of the UM patients [29].

In this study, we show that ctDNA levels early during therapy can predict UM responses to PKCi-based targeted therapy. Interestingly, pre-treatment ctDNA did not predict response or prolonged PFS in our UM patient cohort, even though baseline ctDNA was positively correlated with disease burden and LDH. Importantly, although all responding patients with detectable baseline ctDNA showed reduced levels of ctDNA at EDT, the decrease from PRE to EDT was not significant, and ctDNA at EDT was also reduced in seven of 10 patients showing no clinical benefit to PKCi. Thus, it is the absolute level of EDT ctDNA that is indicative of treatment response in this cohort and we reported similar findings in advanced melanoma patients treated with anti-PD1-based therapy [20]. Considering the level of ctDNA is reflective of both tumor size and metabolic tumor burden [30], it is not surprising that low ctDNA early during therapy would predict treatment response. Our study also confirms that increasing UM ctDNA preceded radiological progression and did so with lead-time ranging from 4 to 10 weeks. A previous study also reported that increasing ctDNA also precedes radiologic detection of UM liver metastases [22]. Thus, the inclusion of ctDNA analysis in Phase I UM trials can provide meaningful data on patients failing to respond to novel therapies, and this may accelerate CT-based confirmation of progression or contribute to adaptive trial design, allowing for earlier access to alternate drugs.

We also utilized a custom targeted NGS panel designed for the detection of 90% of known CM gene mutations and 95% of known UM gene mutations [24]. We confirmed that the allele frequency of driver mutations identified using targeted Ion Torrent NGS was comparable to the mutant allele frequencies determined by ddPCR, although the sensitivity of NGS did not match ddPCR, and the driver GNAQ mutation was not detected by NGS in one patient, presumably due to the low volume of disease. Nevertheless, NGS was able to identify additional mutated genes (i.e., SF3B1, TP53), which have been implicated in UM prognosis and treatment response. For instance, TP53 mutations were detected in the circulation of 5/6 UM patients treated with PKCi+HDM2i. These TP53 mutations may have been selected or expanded in response to therapy as they were not detected at baseline in four patients. As these mutations were detected at low frequency they may represent clonal expansion of tumor cells or hematopoietic stem cells during the process of clonal hematopoiesis. Considering that TP53 loss confers HDM2i resistance [31] and that the TP53 mutations detected in this study are established loss-of-function alterations, ctDNA may prove valuable in the early detection of treatment resistance mechanisms. It is also worth noting that a recent study confirmed that TP53 is significantly disrupted in UM with 11/103 UM showing genomic loss or mutations affecting the TP53 gene [14].

This study was limited by the small sample size and the fact that patients were treated with two distinct treatments based on PKC inhibition. A larger patient sample in a prospective study is required to evaluate the predictive value of baseline ctDNA level and more importantly test the value of including ctDNA as a routine monitoring tool in UM clinical trials.

## 5. Conclusions

In summary, baseline ctDNA in metastatic UM strongly correlates with baseline LDH level and disease volume. Treatment-induced changes in ctDNA and low levels of ctDNA EDT predicted response to PKCi-based targeted therapy and the inclusion of targeted NGS yielded valuable and accurate data about driver mutation frequency and the selection of potential resistance effectors.

Despite proof of concept that ctDNA is a useful biomarker for monitoring response to therapy, in the form of evolution of resistance, and should be included in metastatic UM clinical trials, the most important challenge remains the identification of effective, durable therapies for UM.

## Figures and Tables

**Figure 1 cancers-13-01740-f001:**
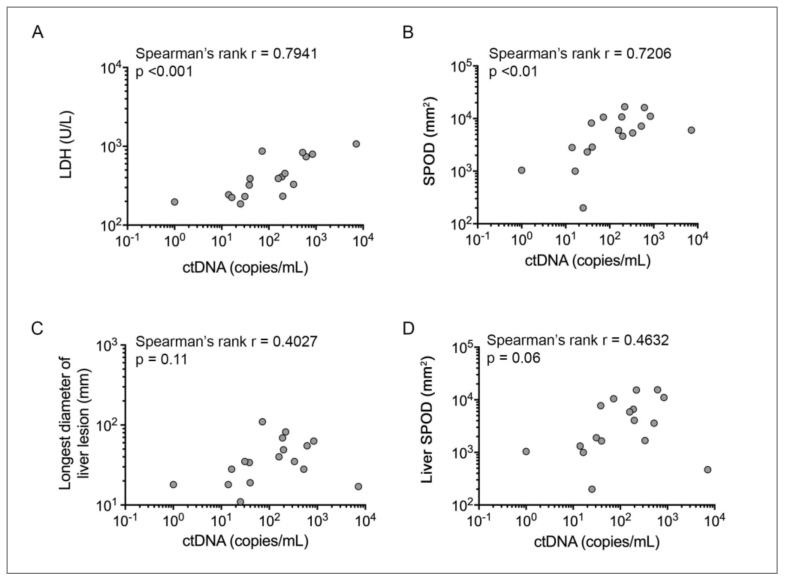
Relationship between uveal melanoma ctDNA (copies/mL), tumor burden and LDH. Spearman’s rank correlation between ctDNA copies/mL and (**A**) LDH (U/L), *p* < 0.001, (**B**) SPOD (mm^2^), *p* < 0.01, (**C**) Longest liver lesion (mm), *p* = 0.11, (**D**) Liver SPOD (mm2), *p* = 0.06. Graph shows ctDNA+1 data.

**Figure 2 cancers-13-01740-f002:**
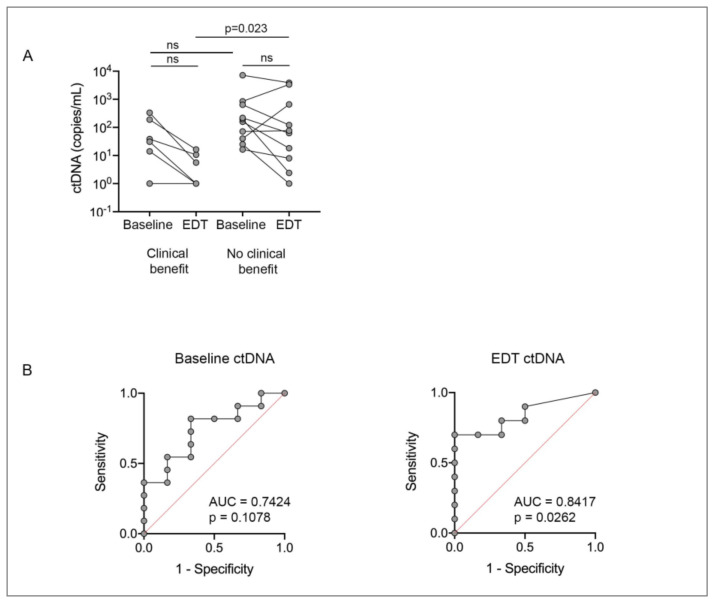
Predictive performance of ctDNA. (**A**) ctDNA changes from baseline to EDT in clinical benefit group (*n* = 6) and no clinical benefit group (*n* = 10) patients. Patient matched PRE-EDT ctDNA levels were compared using Wilcoxon matched-pairs signed rank test, and unpaired PRE or EDT ctDNA levels between clinical benefit and no clinical benefit patients were compared using the Mann–Whitney test. (**B**) ROC curve analysis determined a negative predictive cut-off value (i.e., value providing maximum sensitivity and specificity) for ctDNA > 16.35 copies/mL at EDT for no clinical benefit. ns, not significant; AUC, area under the curve.

**Figure 3 cancers-13-01740-f003:**
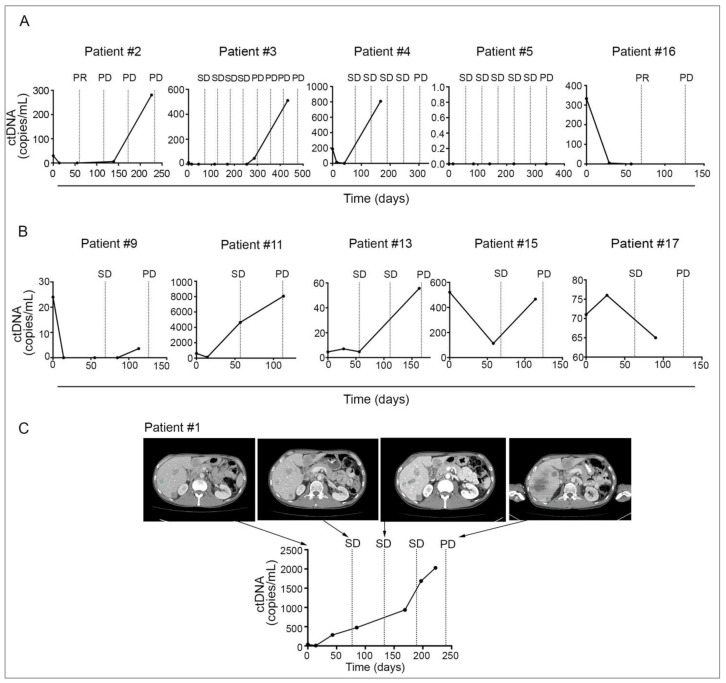
Monitoring of ctDNA in patients treated with PKCi in metastatic UM. ctDNA levels were collected longitudinally during treatment and correlated to CT imaging during baseline, whilst on treatment and on progression. Longitudinal ctDNA monitoring is shown for (**A**) clinical benefit patients, (**B**) no clinical benefit patients and (**C**) CT images and corresponding ctDNA data are shown for clinical benefit patient #1. Only patients #2, #3, #4, #5, #9 and #16 had undetectable ctDNA for the driver oncogene in at least one on-therapy plasma sample. SD, stable disease; PR, partial response; PD, progressive disease.

**Figure 4 cancers-13-01740-f004:**
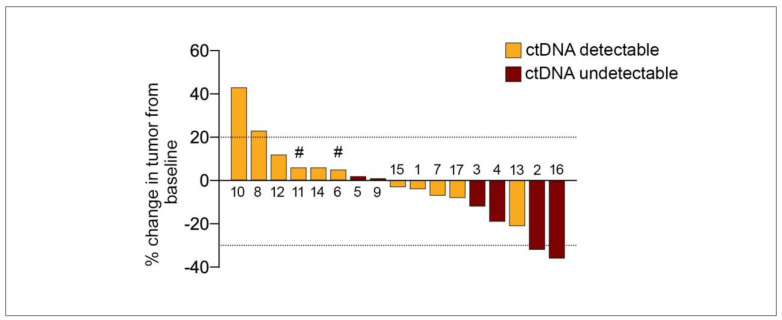
Treatment response in target lesions and ctDNA detectability in UM patients treated with PKCi. Percentage change in target lesions as per RECIST1.1 from 17 patients. Bars are aligned according to decreasing percentage in the sum of target lesions. Positive bars show growth in target lesions and negative bars indicate shrinkage. The dotted line corresponds to a 20% increase or 30% reduction in size of the target lesions. Patients were classified as ctDNA undetectable if at least one on-therapy plasma sample was undetectable for the driver oncogene. Patient IDs are shown above or below bars. # progression of disease with new non-target lesions.

**Table 1 cancers-13-01740-t001:** Baseline clinicopathologic characteristics of uveal melanoma patients.

Characteristics	Patients (*n* = 17)
Age, Median (range)	56 (45–73)
Sex, *n* (%)	
Male	10 (59%)
Female	7 (41%)
ECOG PS, *n* (%)	
0	13 (76%)
≥1	4 (24%)
Mutation, *n* (%)	
GNAQ^Q209P^	6 (35%)
GNA11^Q209L^	8 (47%)
GNAQ^R183Q^	2 (12%)
CYSTLR2^L129Q^	1 (6%)
Number of organs involved by metastatic disease, *n* (%)	
1	3 (17%)
>1	14 (83%)
Liver metastases, *n* (%)	17 (100%)
LDH, *n* (%)	
≤ULN	6 (35%)
>ULN	11 (65%)
Prior Systemic Treatment ^a^	
Yes	13 (76%)
No	4 (24%)
Primary Tumor Type	
Choroidal	12 (70%)
Iris	1 (6%)
Unknown	4 (24%)
Treatment	
PKCi alone	11 (65%)
PKCi + HDM2i	6 (35%)
Best Response ^b^, *n* (%)	
PR	2 (12%)
SD ≥ 6 months	4 (24%)
SD < 6 months	7 (41%)
PD	4 (23%)
PFS (months), median (range)	3.8 (1.7–13.1)
Number of liver lesions, median (range)	9 (1–49)
Liver SPOD (mm^2^), median (range)	3595 (200–15,525)
SPOD (mm^2^), median (range)	5986 (200–16,782)
Largest diameter of liver lesion (mm), median (range)	35 (11–110)

^a^ Prior systemic treatment includes chemotherapy or immunotherapy. ^b^ Patients were stratified into response groups based on RECIST 1.1. Clinical benefit was defined by patients who had partial response (PR) or stable disease (SD) for ≥ 6 months. Patients with SD < 6 months or PD were classified as receiving no clinical benefit. Abbreviations: LDH, lactate dehydrogenase; ULN, upper limit of normal; PR, partial response; SD, stable disease; PD, progressive disease; SPOD, sum of the product of bi-dimensional diameters; PFS, progression-free survival; PKCi, protein kinase C inhibitor (LXS196); HDM2i, HDM2 inhibitor (HDM201); ECOG PS, Eastern Cooperative Oncology Group performance status.

**Table 2 cancers-13-01740-t002:** Liquid biopsy mutation analysis using Ion Torrent next generation sequencing.

Patient ID	Baseline Mutation (MAF %, LOD %)	On-Treatment Mutation (MAF %, LOD%)	Time from Baseline to on-Treatment Sample (Months)
1	GNA11^Q209L^ → (0.7, 0.6)	GNA11^Q209L^ → (23.4, 0.3) TP53^R248Q^ → (23.3, 0.3) TP53^R342^ * → (15.7, 0.3)	8.2
2	GNA11^Q209L^ → (0.5, 0.3)	GNA11^Q209L^ → (9.9, 0.6)	8.5
3	GNA11^Q209L^ → (3.0, 0.3) SF3B1^R625H^ → (1.3, 0.2)	GNA11^Q209L^ → (2.0, 0.3) SF3B1^R625H^ → (1.8, 0.3)	10.0
4	GNAQ^R183H^ → (8.5, 0.2)	GNAQ^R183H^ → (22.4, 0.2)	6.0
5	ND	GNA11^Q209L^ → (4.8, 0.6)	11.3
6	GNAQ^Q209P^ → (32.3, 0.2) SF3B1^R625C^ → (21.2, 0.1)	GNAQ^Q209P^ → (25.4, 0.2) SF3B1^R625C^ → (12.8, 0.2)	0.9
7	GNA11^Q209L^ → (22.7, 0.1) SF3B1^R625L^ → (24.0, 0.1)	GNA11^Q209L^ → (20.9, 0.2) SF3B1^R625L^ → (20.5, 0.2)	3.9
8	GNAQ^Q209P^ → (4.4, 0.1)	GNAQ^Q209P^ → (0.3, 0.2)	1.0
9	ND	ND	3.8
10	CYSLTR2^L129Q^ → (8.1, 0.3)	CYSLTR2^L129Q^ → (0.5, 0.1)	0.9
11	GNA11^Q209L^ → (13.3, 0.1)	GNA11^Q209L^ → (29.5, 0.2) TP53^G244D^ → (0.3, 0.2)	4.0
12	GNA11^Q209L^ → (3.4, 0.3) TP53^Y220C^ → (0.7, 0.3) TP53^R248P^ → (0.3, 0.3)	GNA11^Q209L^ → (14.1, 0.4)	0.9
13	GNAQ^Q209P^ → (0.8, 0.2)	GNAQ^Q209P^ → (1.0, 0.2) TP53^R248G^ → (0.3, 0.2)	5.4
14	GNAQ^Q209P^ → (6.3, 0.2) SF3B1^R625H^ → (8.6, 0.2)	GNAQ^Q209P^ → (3.1, 0.4) SF3B1^R625H^ → (6.1, 0.3)	3.8
15	GNAQ^Q209P^ → (20.1, 0.2)	GNAQ^Q209P^ → (9.7, 0.2)	3.8
16	GNAQ^Q209P^ → (5.9, 0.2)	ND	2.4
17	GNAQ^R183Q^ → (4.2, 0.2)	GNAQ^R183Q^ → (11.6, 0.3) TP53^S215G^ → (0.4, 0.3)	3.4

ND, not detected; MAF, mutant allele frequency; LOD, limit of detection. Timing of the on-treatment plasma sample collection is also shown. *, indicates premature termination codon. All mutations shown had MAF > LOD.

## Data Availability

The next generation sequencing and ddPCR data presented in this study are available on request from the corresponding author. The data are not publicly available due to patient confidentiality.

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
