# Peer review of "Circulating Tumor DNA Reflects Uveal Melanoma Responses to Protein Kinase C Inhibition"

_cancers, 2021, doi:10.3390/cancers13071740_

Round 1

Reviewer 1 Report

Park JJ et al examined circulating tumor DNA (crDNA) in 17 metastatic UM patients treated with the PKC inhibitor by using digital droplet PCR and next generation sequencing. They found baseline ctDNA strongly correlated with baseline LDH and base line tumor burden. Early during treatment (EDT) ctDNA accurately predicted patients with clinical benefit to PKCi. Furthermore, they found that ctDNA value is informative to detect disease progression in patients with targeted therapy. Finally, targeted NGS of ctDNA successfully identified both driver mutations and some novel mutations in patients resistance to PKCi therapy. They concluded that the inclusion of longitudinal ctDNA monitoring in UM can be beneficial to clinical trial design. This study is very clinically relevant and has important clinical implication. However, the main concern is the patient numbers are too small, thus possibly leading to the premature conclusion. My other comments are below:

  1. The authors identified TP53 mutation in some patients with PKCi/HDM201 treatment. They suggest that this TP53 mutation possibly results in resistance to treatment. This conclusion is premature. The reasons are: (1) both patient 1 and patient 16 that are response to treatment have also TP53 mutation after treatment; (2) It is very difficult to know whether the TP53 mutation is due to selection by drug treatment or undetectable in pre-treatment samples; (3) No evidence support TP53 mutation lead to resistance to PKCi/HDM201 treatment.
  2. In Fig.2, the baseline ctDNA difference between clinical benefit group and no clinical benefit group is not statistically significant. It is unclear how EDT ctDNA can be used as a prognostic factor. What the threshold of EDT ctDNA amount should decide whether patients are response to treatment or not?
  3. 2B is hard to understand. Please give more detail. The Font is too small in Fig.3.
  4. Because Bap1 mutation is found in many metastatic UMs, please discuss why no Bap1 mutation is found in all 17 patients.
  5. There are many typo errors in the text. For example, in line 35, there should be a “.” after (NGS). In line 129, “targed” should be “targeted”. In line “332”, “utilised” should be “utilized”.

Author Response

We appreciate the thoughtful comments of reviewer 1 and have addressed the comments below:

1. The authors identified TP53 mutation in some patients with PKCi/HDM201 treatment. They suggest that this TP53 mutation possibly results in resistance to treatment. This conclusion is premature. The reasons are: (1) both patient 1 and patient 16 that are response to treatment have also TP53 mutation after treatment; (2) It is very difficult to know whether the TP53 mutation is due to selection by drug treatment or undetectable in pre-treatment samples; (3) No evidence support TP53 mutation lead to resistance to PKCi/HDM201 treatment.

We agree with the reviewer that it is difficult to know whether the identification of TP53 mutations is due to selection or expansion of pre-existing mutations – although both are important resistance mechanisms. We were very careful with our language throughout the manuscript. In the results we provided the data and stated the following:

Line 273-276:‘Interestingly, in 4 of 5 PKCi+HDM2i patients the TP53 variants were not identified pre-treatment, suggesting the possibility of selection during treatment. It is important to note that many TP53 mutations were detected at low allele frequencies that were below the 0.2% limit of detection of our NGS assay(24) and thus would require further validation.’

We were also careful to highlight limitations in the results section as follows:

Line 316-318: ‘As these mutations were detected at low frequency they may represent clonal expansion of tumor cells or hematopoietic stem cells during the process of clonal hematopoiesis.’

The comment by the reviewer that the TP53 mutations may have been present pre-treatment is valid, although the fact they were detectable after treatment suggest selection and/or expansion – and we have amended the results section to the following:

Line 315-318: ‘These TP53 mutations may have been selected or expanded in response to therapy as they were not detected at baseline in four patients’

In terms of TP53 mutation leading to resistance to combination PKCi/HDM201 – our data are the first to suggest this as a possibility, – but it is pertinent that TP53 mutations confer resistance to HDM201, as indicated in the results and citation provided. We have also amended the Abstract to indicate these mechanisms are putative.

Line 42: targeted NGS of ctDNA revealed putative resistance mechanisms prior to radiological progression

Line 318: ‘Considering that TP53 loss confers HDM2i resistance(31)’

2. In Fig.2, the baseline ctDNA difference between clinical benefit group and no clinical benefit group is not statistically significant. It is unclear how EDT ctDNA can be used as a prognostic factor. What the threshold of EDT ctDNA amount should decide whether patients are response to treatment or not?

The reviewer raises an important limitation of ctDNA in melanoma – i.e the baseline ctDNA is not always prognostic in melanoma (see Lee et al. Ann Oncol 28:1130). Nevertheless, there is value in the EDT samples –EDT sample analysis enables rapid and real-time assessment of response to novel therapies. In our study ctDNA increases preceded radiological progression with a lead time of 4-10 weeks (described in Line 304-305).We have also provided the threshold data on the EDT ctDNA in the results:

Line 213-215: Based on ROC curve analysis the sensitivity, specificity for ctDNA >16.35 copies/mL at EDT for no clinical benefit was 70% and 100%. The positive and negative predictive for ctDNA >16.35 copies/mL were 100% and 67%, respectively.

3. 2B is hard to understand. Please give more detail. The Font is too small in Fig.3.

The following additional description is provided in Figure 2B legend

Line 220-222: B) ROC curve analysis determined a negative predictive cut-off value (i.e value providing maximum sensitivity and specificity) for ctDNA >16.35 copies/mL at EDT for no clinical benefit. ns, not significant; AUC, area under the curve

The font size in Figure 3 graphs has been increased

4. Because Bap1 mutation is found in many metastatic UMs, please discuss why no Bap1 mutation is found in all 17 patients.

The Ion Ampliseq HD melanoma panel we used in this report does not include the BAP1 gene. We have clarified this in the method section 2.6

Line 124:  This melanoma gene panel does not cover the BAP1 gene

There are many typo errors in the text. For example, in line 35, there should be a “.” after (NGS). In line 129, “targed” should be “targeted”. In line “332”, “utilised” should be “utilized”.

We have conducted a thorough grammar and spell check on this manuscript

Reviewer 2 Report

The authors performed additional ctDNA analyses in 17 uveal melanoma patients of a phase I study. The results are presented in detail, but there are inconsistencies in some parts, probably due to errors. I cannot share the central statement based on the results presented (“In this study, we show that ctDNA levels early during therapy can predict UM responses to PKCi-based targeted therapy, with consistent and significant reductions in ctDNA EDT in responding patients.”)

Here are my comments in detail:

The authors define clinical benefit as follows: “Clinical benefit was defined by patients who had partial response (PR) or stable disease (SD) for > 6 months.” At least 6 months specifically means at least 180 days. So I don't understand how you could describe “PD” in Figure 3 A (clinical benefit patients) in patient #2 and #16 at day 100 and 125, respectively. According to your definition they should be classified to the “no clinical benefit group.

The authors classify only in > or < 6 monthly intervals. It would be better to classify in either ≤ or > / < or ≥ 6 monthly intervals.

Regarding the method section, it should be explained, why only in 17 patients, ctDNA monitoring was performed. The study (according to https://clinicaltrials.gov/ct2/show/NCT02601378) intended to recruit 68 patients. What were the criteria for additional ctDNA monitoring? What about the remaining patients?

The authors specify the timing of the ctDNA samples: “Plasma samples were collected at baseline (prior to therapy start), EDT (early during treatment between 14 –30 days of commencing PKCi-based therapy) and at later time points during therapy. PROG samples were defined as plasma samples taken within 30 days (before or after) of disease progression confirmed by imaging or clinical progression as determined by the treating clinician. “ Please explain exactly, what time point is displayed in table 2 “on-treatment mutation” (0.9-11.3 months from baseline). In patient #1 and #2 it seems to be the last ctDNA sample, but in patient #3 it is not the last but the pre-last one….

What are the criteria for the selection of the time point “on-treatment”? This must be added to the methods part. Why did you not mention copies/ml?  

EDT is defined as 14-30 days of commencing PKCi-based therapy. In view to a targeted therapy approach it seems to me to be relevant whether the ctDNA control is done on day 14 or day 30, the more as the dose of therapy might have been modified (phase 1 trial). Here it must be mentioned whether the patients all had the same dose of PKCi-based therapy at the time of the EDT control. Furthermore, it should be considered if patients had treatment interruptions for example due to adverse-events. If so, increasing of ctDNA could be also due to dose interruptions.

The authors display several patients achieving undetectable ctDNA levels during the course of their therapy. At least 8 patients can be counted according to figure 3 (patient #2,#3,#4,#5,#16,#9,#11,#1). In contrast you display in figure 4 only 6 patients with “ctDNA undetectable”. The corresponding figure legend: “Patients were classified as ctDNA undetectable if at least one on-therapy plasma sample was undetectable for the driver oncogen“. It is confusing to me / the manuscript seems to contain some errors...

Please correct also: In 3.1 the authors write “On commencement of the treatment, 11 (65%) patients had elevated LDH levels” but the table displays 11 patients with normal LDH values. This must be corrected.

The authors write “EDT ctDNA, but notPRE ctDNA or change from PRE to EDT, accurately predicted clinical benefit to PKCi based therapy (AUC 0.84, [95% confidence interval, 0.65 –1.0, p=0.026]) (Figure 2B).” This sentence needs to be discussed more intensively, please refer also to the following comments:

I do not agree with the authors` statement: “In this study, we show that ctDNA levels early during therapy can predict UM responses to PKCi-based targeted therapy, with consistent and significant reductions in ctDNA EDT in responding patients.” In figure 2 we can see that almost all of the patients (n=13) had a significant reduction in ctDNA EDT. 6 of them had clinical benefit but there are also 7 patients with no clinical benefit. Therefore, it is not correct to say that the reduction indicates a response. The fact that despite a visible reduction in ctDNA (in some cases even more pronounced than in the clinical benefit patients), progression nevertheless occurred must be discussed in detail. Currently, possible explanations are missing. We cannot say from the results of this study that a significant reduction in ctDNA EDT can predict the response to PKI.

The authors themselves write that neither baseline ctDNA levels nor changes from PRE to EDT are not suitable for predicting response (although it correlates with tumor burden and LDH). Nevertheless they calculated (how?) a cut-off (ctDNA > 16.35 copies/mL at EDT) for the no benefit group…“The sensitivity, specificity, and positive and negative predictive value for ctDNA > 16.35 copies/mL at EDT for no clinical benefit was 73%, 100%, 100% and 67%, respectively.”

They must explain, how sensitivity, specificity, positive and negative predictive value and the cut-off (16.35 copies/mL) were calculated. This has to be added to the methods part.

Thus, the central statement would be that ctDNA level > 16.35 copies/mL at EDT is predictive of non-response to PKCi-based targeted therapy.

The number of cases is very small and in addition, the therapy regimen (and probably also the doses) were different. It would be absolutely important to evaluate the remaining patients of this study and to check again in a larger cohort whether the reduction between baseline and EDT or at least the baseline value could be used as predictive markers. 

The results of this study do not convince me. On the one hand, because of the errors most likely included in important places, the small number of cases, and the inconclusive interpretation of the data.

Author Response

We thank the reviewer for their thoughtful comments and have addressed the comments below:

1. The authors define clinical benefit as follows: “Clinical benefit was defined by patients who had partial response (PR) or stable disease (SD) for > 6 months.” At least 6 months specifically means at least 180 days. So I don't understand how you could describe “PD” in Figure 3 A (clinical benefit patients) in patient #2 and #16 at day 100 and 125, respectively. According to your definition they should be classified to the “no clinical benefit group".

Patients #2 and #16 had partial response (PR) to therapy hence considered in clinical benefit group.

2. The authors classify only in > or < 6 monthly intervals. It would be better to classify in either ≤ or > / < or ≥ 6 monthly intervals.

We have changed to  ≥ 6 months from > 6 months.

3. Regarding the method section, it should be explained, why only in 17 patients, ctDNA monitoring was performed. The study (according to https://clinicaltrials.gov/ct2/show/NCT02601378) intended to recruit 68 patients. What were the criteria for additional ctDNA monitoring? What about the remaining patients?

The clinical trial was based on multiple international clinical trial sites. This study analysing ctDNA is from 17 patients participating in the clinical trial from a single institution which consistently banked the plasma specimens. This was indicated in Section 2.1 :

Line 78-79: 'Seventeen patients with metastatic UM with known mutations in GNAQ, GNA11 and CYSTLR2, treated with the novel PKCi, LXS196 (n = 17) at Westmead Hospital, Sydney, Australia'

4. The authors specify the timing of the ctDNA samples: “Plasma samples were collected at baseline (prior to therapy start), EDT (early during treatment between 14 –30 days of commencing PKCi-based therapy) and at later time points during therapy. PROG samples were defined as plasma samples taken within 30 days (before or after) of disease progression confirmed by imaging or clinical progression as determined by the treating clinician. “ Please explain exactly, what time point is displayed in table 2 “on-treatment mutation” (0.9-11.3 months from baseline). In patient #1 and #2 it seems to be the last ctDNA sample, but in patient #3 it is not the last but the pre-last one….

We have updated the description of plasma samples in section 2.3 as shown below. It is important to note that the last available on-treatment plasma sample was used for NGS analysis, and thus we provide the timing of on-treatment sample collection in Table 2.

Line 96-100: Plasma samples were collected at baseline (prior to therapy start), EDT (early during treatment between 14 – 30 days of commencing PKCi-based therapy) and at later time points during therapy (on-treatment samples). PROG samples were defined as plasma samples taken within 30 days (before or after) of disease progression confirmed by imaging or clinical progression as determined by the treating clinician. NGS analysis was performed on baseline plasma samples and on the last available on-treatment plasma sample.

We also added an addition footnote in Table 2 to clarify the on-treatment sample collection.

Line 279-280: Timing of the on-treatment plasma sample collection is also shown. 

5. What are the criteria for the selection of the time point “on-treatment”? This must be added to the methods part. Why did you not mention copies/ml?  

On treatment sample details provided in Line 96-100 as described above.

ddPCR results are often reported in copies/mL, and this has been added to methods section 2.5

Line 118-119: ddPCR results in our study is reported in copies/mL.

6. EDT is defined as 14-30 days of commencing PKCi-based therapy. In view to a targeted therapy approach it seems to me to be relevant whether the ctDNA control is done on day 14 or day 30, the more as the dose of therapy might have been modified (phase 1 trial). Here it must be mentioned whether the patients all had the same dose of PKCi-based therapy at the time of the EDT control. Furthermore, it should be considered if patients had treatment interruptions for example due to adverse-events. If so, increasing of ctDNA could be also due to dose interruptions.

Due to the timing of blood collection during this trial, we broadened the definition of EDT to include 14 – 30 days from commencing PKCi- based therapy. As this was a phase 1 dose escalation clinical trial, PKCi dosing did vary and we have included these details in the updated Supplementary Table 1. 10/16 patients had EDT sample on day 14, 1 patient each had EDT sample on day 15, 26, 27, 28, 29, 30. Only Patient # 17 had a 3 day of dose interruption during PRE-EDT sample period and this has been detailed as a footnote in Supplementary Table 1. Importantly, the EDT sample derived from Patient #17 was collected while on therapy.

aPatient #17 had a 3 day does interruption during the PRE – EDT sample period, and was on treatment at the time of EDT sample collection.

7. The authors display several patients achieving undetectable ctDNA levels during the course of their therapy. At least 8 patients can be counted according to figure 3 (patient #2,#3,#4,#5,#16,#9,#11,#1). In contrast you display in figure 4 only 6 patients with “ctDNA undetectable”. The corresponding figure legend: “Patients were classified as ctDNA undetectable if at least one on-therapy plasma sample was undetectable for the driver oncogen“. It is confusing to me / the manuscript seems to contain some errors...

Only 6 patients (#2, #3, #4, #5, #9, #16) as displayed on Figure 4 had undetectable ctDNA. In Figure 3, due to the scale of the y-axis , Patients #1 and #11, appears to have ctDNA undetectable, however #1 ctDNA NADIR is 9.6 copies/mL and #11 ctDNA NADIR is 1.4 copies/mL and did not become undetectable.

We have added the following sentence in the Figure 3 legend to improve clarity:
Line 242-243: Only patients #2, #3, #4, #5, #9 and #16 had undetectable ctDNA for the driver oncogene in at least one on-therapy plasma sample.

8. Please correct also: In 3.1 the authors write “On commencement of the treatment, 11 (65%) patients had elevated LDH levels” but the table displays 11 patients with normal LDH values. This must be corrected.

Table 1 has been corrected with 11 (65%) patients having >ULN LDH level.

9. I do not agree with the authors` statement: “In this study, we show that ctDNA levels early during therapy can predict UM responses to PKCi-based targeted therapy, with consistent and significant reductions in ctDNA EDT in responding patients.” In figure 2 we can see that almost all of the patients (n=13) had a significant reduction in ctDNA EDT. 6 of them had clinical benefit but there are also 7 patients with no clinical benefit. Therefore, it is not correct to say that the reduction indicates a response. The fact that despite a visible reduction in ctDNA (in some cases even more pronounced than in the clinical benefit patients), progression nevertheless occurred must be discussed in detail. Currently, possible explanations are missing. We cannot say from the results of this study that a significant reduction in ctDNA EDT can predict the response to PKI.

The reviewer is correct that the change from PRE to EDT did not predict clinical benefit – as shown in Figure 2. In fact, this is what we have concluded in the results:

Line 210-212: The predictive accuracy of ctDNA was also examined using receiver operator characteristic classification (ROC) curves. EDT ctDNA, but not PRE ctDNA or change from PRE to EDT, accurately predicted clinical benefit to PKCi based therapy (AUC 0.84, [95% confidence interval, 0.65 – 1.0, p=0.026]) (Figure 2B).

We have expanded our discussion to describe the implications of these findings:

Line 298-303: Importantly, although all responding patients with detectable baseline ctDNA showed reduced levels of ctDNA at EDT, the decrease from PRE to EDT was not significant, and ctDNA at EDT was also reduced in 7 of 10 patients showing no clinical benefit to PKCi. Thus, it is the absolute level of EDT ctDNA that is indicative of treatment response in this cohort and we reported similar findings in advanced melanoma patients treated with anti-PD1-based therapy(20). Considering the level of ctDNA is reflective of both tumour size and metabolic tumour burden(30), it is not surprising that low ctDNA early during therapy would predict treatment response.

The statement in the discussion that is highlighted by the reviewer has also been amended to improve accuracy

Line 296: In this study, we show that ctDNA levels early during therapy can predict UM responses to PKCi-based targeted therapy.

10. They must explain, how sensitivity, specificity, positive and negative predictive value and the cut-off (16.35 copies/mL) were calculated. This has to be added to the methods part.

We have included the positive and negative predictive value calculations in the methods section 2.7: Statistical analysis.

Line 133-137: Positive predictive value for EDT >16.35 copies/mL was calculated using the following formula: Number of patients showing no clinical benefit with EDT ctDNA >16.35 copies/mL divided by number of patients with EDT ctDNA >16.35 copies/mL. Negative predictive value was determined as follows: Number of responding patients with EDT ctDNA ≤16.35 copies/mL divided by number of patients with EDT ctDNA ≤16.35 copies/mL.

We have also specified that the threshold of >16.35 copies/mL was determined via the ROC curve analysis in the results.

Line 213-215: Based on ROC curve analysis the sensitivity, specificity for ctDNA >16.35 copies/mL at EDT for no clinical benefit was 70% and 100%. The positive and negative predictive for ctDNA >16.35 copies/mL were 100% and 67%, respectively.

11. The results of this study do not convince me. On the one hand, because of the errors most likely included in important places, the small number of cases, and the inconclusive interpretation of the data.

The number of patients in this study is relatively small, however UM is a rare cancer with low incidence and is difficult to conduct ctDNA studies due to poor prognosis and relatively shorter PFS. It is important to note that this study has the largest cohort of patients to analyse ctDNA as a biomarker treated with a targeted therapy (PKCi) and believe strongly this study has significant clinical relevance.  We respectfully disagree regarding errors in important places – but we have improved clarity throughout based on all the comments provided by the reviewers.

Reviewer 3 Report

In “Circulating Tumor DNA reflects Uveal Melanoma responses to Protein Kinase C inhibition” Park et al. found significant correlation between reduction of level of EDT ctDNA and positive response to PKC inhibitor targeted therapy and also confirmed other results of prior reports concerning the ctDNA in UM.

The study was well designed and the paper is well written. The discussion and conclusions are in line with the presented findings and although the number of patients is some limitation of the results, I consider it worth publishing.

Some minor text adjustments seem appropriate.

Author Response

We thank the reviewer for their comments.

We have conducted a thorough grammar and spell check throughout to improve the manuscript. 

Reviewer 4 Report

In the manuscript “Circulating Tumor DNA reflects Uveal Melanoma responses to Protein Kinase C inhibition" by John J. Park et al., the authors conduct an innovative study on the importance of circulating tumour DNA on the diagnosis, monitoring and prognosis of uveal melanoma patients. Even though the study involves a small cohort uveal melanoma patients, it clearly shows that baseline ctDNA in metastatic uveal melanomas has a strong correlation with the disease volume and baseline levels of LDH. In addition, the ctDNA information was also able to help predicting the response to PKC inhibitor-based targeted therapy.

Therefore, I consider that this study is innovative, relevant, and impactful and meets the standards to be published in Cancers in the present format.

Author Response

We appreciate the positive comments provided by this reviewer.

Round 2

Reviewer 1 Report

The authors have addressed my concerns.